# Responsible AI Considerations in Text Summarization Research: A Review of Current Practices

**Yu Lu Liu**[1,2]   **Meng Cao**[1,2]   **Su Lin Blodgett**[3]
**Jackie Chi Kit Cheung**[1,2,4]   **Alexandra Olteanu**[3]   **Adam Trischler**[3]
[1]Mila – Quebec Artificial Intelligence Institute   [2]McGill University
[3]Microsoft Research, Montréal, Canada   [4]Canada CIFAR AI Chair
{yu.l.liu, meng.cao}@mail.mcgill.ca  jackie.cheung@mcgill.ca
{sulin.blodgett, alexandra.olteanu}@microsoft.com
adam.trischler@gmail.com

## Abstract

AI and NLP publication venues have increasingly encouraged researchers to reflect on possible ethical considerations, adverse impacts, and other responsible AI issues their work might engender. However, for specific NLP tasks our understanding of how prevalent such issues are, or when and why these issues are likely to arise, remains limited. Focusing on text summarization—a common NLP task largely overlooked by the responsible AI community—we examine research and reporting practices in the current literature. We conduct a multi-round qualitative analysis of 333 summarization papers from the ACL Anthology published between 2020–2022. We focus on how, which, and when responsible AI issues are covered, which relevant stakeholders are considered, and mismatches between stated and realized research goals. We also discuss current evaluation practices and consider how authors discuss the limitations of both prior work and their own work. Overall, we find that relatively few papers engage with possible stakeholders or contexts of use, which limits their consideration of potential downstream adverse impacts or other responsible AI issues. Based on our findings, we make recommendations on concrete practices and research directions.

## 1 Introduction

Text summarization is an important NLP task where the goal is to generate a shorter version of an input text while preserving its main ideas. Applications involving text summarization range from storyline generation and sentence compression to meeting notes summarization and email commitment reminders.

As their capabilities increase, especially with the emergence of large language models, automatic text summarization systems have seen increasing use—despite the known risks of generating incorrect, biased, or otherwise harmful summaries.

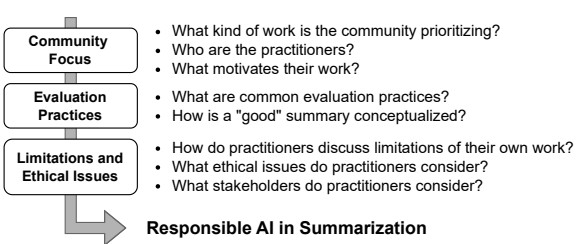

Figure 1: Overview of questions we examine when analyzing practices related to how the contemporary text summarization literature engages with RAI issues.

Generated summaries might, e.g., misgender the people they describe; give rise to libelous representations by failing to appropriately qualify claims; mislead users by giving rise to inferences that are ambiguous or unsupported by the source text; represent contested topics unfairly; or be susceptible to adversarial perturbations in the source text.

Recently, there have been growing efforts to incorporate, in AI and NLP research practice, reflections about ethical considerations, adverse impacts, and other responsible AI (RAI) issues that NLP and AI research—and related applications—can exacerbate (Boyarskaya et al., 2020; Nanayakkara et al., 2021; Hardmeier et al., 2021). *Despite these efforts and the array of risks, little work has comprehensively examined responsible AI concerns arising from summarization systems.*

In this work, we investigate research and reporting practices related to how, when, and which RAI issues are covered in the contemporary text summarization literature. To examine these practices, we developed, through a multi-round annotation process, a set of annotation guidelines targeting aspects relevant to RAI. Following these guidelines, we conducted a detailed, systematic review of 333 summarization papers published between 2020 and 2022 in the ACL Anthology.[1] Specifically, we examine how authors discuss limitations of both prior work and their own work,

---

[1]Paper annotations are available upon request.

which RAI issues they consider, the relevant stakeholders they imagine or serve, as well as how stated and realized research goals might often differ.

We do so to help foreground our choices as a community—about how we write, how we frame problems, how we consider social context, and how we broadly think about RAI issues—and to make these choices explicit (rather than implicit) so that we may better understand their implications. Since the NLP community has only recently started to prioritize these issues, taking an early snapshot of emerging practices can provide insight into why the community might be struggling with considering limitations of its work, ethical considerations, adverse impacts, and other related issues.

We find that despite the introduction of impact statements and ethical considerations sections at both NLP (Benotti and Blackburn, 2022) and AI conferences more generally (Ashurst et al., 2022; Nanayakkara et al., 2021; Boyarskaya et al., 2020), relatively few papers engage with possible stakeholders or contexts of use, and fewer still—less than 15% in the set we reviewed—with responsible AI issues. We discuss how this limits the space of responsible AI issues of which the community may be aware, as well as the community's capacity to speculate effectively on potential issues. We make recommendations for how the community can improve summarization research practices to be more responsible.

## 2 Background & Related Work

**Assessing summaries.** Text summarization is a longstanding NLP research topic with a growing number of applications (Mani, 2001; Nenkova and McKeown, 2012), particularly with the increased availability of both datasets and neural summarizers (Dong, 2018). Text summarization systems are often evaluated using string and word matching metrics like ROUGE (Lin, 2004) to assess the similarity between references or gold summaries and system generated summaries. These methods often do not correlate well with human judgments, spurring research into developing automatic metrics with higher human correlations (Fabbri et al., 2021). Nevertheless, automatic metrics can obfuscate when systems may or may not work; e.g., are the summaries more likely to leave out a certain type of content? Do they work equally well for content by different speakers? When human judgments are included, they often examine intrinsic qualities of the text such as whether the summaries preserve relevant content or are non-redundant, fluent and coherent (Gkatzia and Mahamood, 2015), but rarely extrinsic criteria like how the summaries are used in downstream applications (Zhou et al., 2022).

More recently, a growing emphasis has been placed on ensuring that generated summaries are consistent with the source text, as abstractive systems risk generating so-called "hallucinations," i.e., text that distorts or is unsupported by the source text (Cao et al., 2020; Dong et al., 2020; Falke et al., 2019; Kryscinski et al., 2020; Kumar and Cheung, 2019). Related concerns about factuality, accuracy, and coherency have all been bundled under hallucinations, obscuring what issues the authors are after and the range of harms or adverse impacts they can bring about. Our work examines how assessment practices might limit the space of responsible AI issues the community considers by possibly obfuscating some issues and foregrounding others.

**Responsible AI and summarization.** While a great deal of work on responsible AI issues has emerged for NLP broadly, much less work has addressed summarization specifically. Carenini and Cheung (2008) examine whether a summary reflects the distribution of opinions in source documents. Shandilya et al. (2018) and Dash et al. (2019) consider whether summarization systems fairly represent document authors from different demographic groups, while Shandilya et al. (2020) explore readers' perceptions of fairness in summaries, finding that ROUGE metrics are not well-suited to capturing perceptions of summary fairness. Meanwhile, Keswani and Celis (2021) find that summarization systems produce summaries under-representing already-minoritized language varieties. In our analysis of the summarization literature, we explore to what extent papers acknowledge these existing concerns and aim to uncover issues not previously raised by existing work.

**Meta-analyses in NLP.** We draw inspiration from recent work that analyzes research and reporting practices in NLP. Blodgett et al. (2020) explore how NLP papers describe "bias," finding that definitions are often vague, vary widely, and may not be well-matched to accompanying technical approaches, while Benotti and Blackburn (2022) examine ethical considerations sections in ACL 2021 papers, finding that relatively few (∼15%) include such sections, and that some of these (∼20%) do not

meaningfully address either benefits or harms of the research. Blodgett et al. (2021) examine how four benchmark datasets conceptualize and operationalize stereotyping, while Devinney et al. (2022) analyze papers on gender bias in NLP to uncover how gender is theorized, finding that theorizations rarely are made explicit or engage with gender theories beyond NLP. Via interviews and a survey, Zhou et al. (2022) examine practitioners' assumptions and practices when evaluating natural language generation systems. Elsewhere, work has analyzed evaluation practices in natural language generation (Gkatzia and Mahamood, 2015; van der Lee et al., 2019; Howcroft et al., 2020, i.a.). We draw on these papers in our own investigation of how papers describe the goals of their work, the approaches they take in evaluating progress towards those goals, and the responsible AI issues they may raise.

## 3 Methods

To understand research practices surrounding how, when, and which RAI issues are or should be considered by the text summarization community, we conducted a systematic survey of the summarization literature. To do so, we followed several steps: we first i) gathered a collection of recent text summarization papers to be examined and annotated (§3.1) and ii) reviewed a small set of text summarization papers published in various venues to explore relevant practices (§3.2.1). Drawing on this exploratory review, iii) we then developed an annotation scheme (detailed in §3.3), which we used to annotate our collections of text summarization papers (§3.2.2). Finally, iv) we analyze the annotations to understand emerging practices (§3.2.1).

### 3.1 Paper Collection

We focused on papers published between 2020 and 2022 in the ACL Anthology. To do so, we first gathered all papers with "summarization" in their title or abstract.[2] After manually removing unrelated papers (i.e., papers using "summarization" for purposes other than the task of textual summarization), we obtain 401 summarization papers. We then manually filter out papers where the main focus was not text summarization (e.g., natural language generation papers where summarization is one of many evaluated tasks). This resulted in the set of 333 papers that we annotate.

---

[2]We excluded demonstration, shared task description, tutorial, and workshop papers.

### 3.2 Paper Review & Annotation

#### 3.2.1 Exploratory Review

To scope our literature review and determine the practices we wanted to capture, we started with a small set of 8 summarization papers. We wanted a variety of papers in terms of publication venues and domain, and we were also interested in papers with an "ethical considerations" section. The selected papers are either published at *CL venues (DeYoung et al., 2021; Zhao et al., 2020; Feng et al., 2021; Aralikatte et al., 2021; Zhang et al., 2021b) or HCI and social computing venues (Zhang et al., 2020; Tran et al., 2020; Molenaar et al., 2020), and they cover summarization of medical literature, medical dialogues, legal cases, and emails. We observed differences among these papers, with those published at HCI/social computing venues focusing more on how summarization systems are used and on their stakeholders, which, along with our research questions, informed our early annotation aspects. These aspects included *mentioned stakeholders*, *author affiliation*, *domain*, *limitations*, and more.

#### 3.2.2 Paper Annotation

From this starting point, we developed a common annotation scheme over several iterations (Rounds 1 & 2 below), which was then used to annotate the collection of text summarization papers. Appendix A provides detailed statistics on the process.

**Round 1: Developing & refining the annotation scheme.** Guided by the initial annotation dimensions described above, every author open-coded 20 papers such that each paper was coded by 2 authors, totaling 60 papers. We periodically compared our annotations, updated the annotation scheme to resolve confusions and disagreements, and revised our annotations when necessary. For example, we split the initial *domain* category into *intended domain* and *actual domain* to better track differences between the two, which we noticed in many papers. At the end of this round, we arrived at the scheme overviewed in §3.3.

**Round 2: Applying & clarifying the annotation scheme.** Using the scheme, we coded a larger subset of 131 papers. While in this round each paper was coded by a single author, we continued to periodically discuss ambiguous cases, clarify the annotation scheme, and update annotations accordingly.

**Round 3: Hired annotators.** With the guidelines finalized, for the remainder of the papers, we hired

7 annotators—graduate students in the field of NLP. We paid them at a rate of 30 CAD per hour, which is roughly equivalent to the wage of teaching assistants at our university. We started by briefing the annotators with a 1.5-hour paid training session on our project goal, how their annotations would be used, and the annotation scheme, illustrated by examples from the first two annotation rounds. We then scheduled 26 two-hour sessions held via video-conference, with one author present at all times to answer questions and offer clarifications. The annotators could choose which and how many sessions to attend. The annotators were reminded of their right to periodically suspend or quit the annotation, without any impact on their pay. A total of 142 papers were annotated by hired annotators.

## 3.3 Annotation Scheme

To help us reflect on when RAI issues are brought up, how they are framed, and by whom, our annotation scheme covers aspects related to each paper's goals & authors (§3.3.1), evaluation practices (§3.3.2), as well as stakeholders (if any mentioned), limitations (of prior work or current work), and ethical considerations (§3.3.3).

### 3.3.1 Paper Authors & Goals

As we aim to examine how practitioners engage with RAI in their work, we need to know who the practitioners are, what their work is, and what motivates their work. These aspects not only contextualize our survey, but also provide cues about potential usage scenarios, which may determine what harms are likely to occur. Specifically, we consider the following aspects:

**Contributions:** the type(s) of contribution a paper makes to the research community, including a new *dataset*, a summarization *system* (including new models, methods, or techniques), an evaluation *metric*, an *application* of automatic text summarization, a comprehensive *evaluation* of a collection of existing artifacts, or *other* types of contributions. This allows us to examine, for instance, whether authors of papers with certain types of contributions are more likely to engage in ethical reflection.

**Intended domain:** the domain(s) the work is stated to be developed for, including *news* articles, *dialogue*, computer *code*, *medical* documents, *blogs* (e.g., Twitter), *opinions* (e.g., customer reviews), *scientific* articles, *wiki* (Wikipedia or Wikipedia-like platforms), *other* domains, or *general*. The latter code is used when nothing is explicitly specified throughout the paper's introduction (i.e., a failure to state an intended domain), or when the paper explicitly intends to be general (i.e., explicitly stating that its contribution is general-purpose, or that it can be used in any domain or application).

**Research goals:** authors' stated goals. Annotators either copy or summarize the paper's goal, based on the abstract and the introduction of the paper. This provides additional context to the contributions, intended use or domain, as well as issues with current practices the authors aim to address.

**Affiliation:** the authors' affiliation, including whether there is at least one author affiliated with an *academic* institution, with *industry*, or *other* organizations (e.g., government or NGOs).

### 3.3.2 Data & Evaluation Practices

Evaluation practices reflect the space of concerns (including RAI issues) that the community is aware of, and can also give rise to their own RAI issues. We therefore annotate papers according to:

**Actual domain:** the domain(s) of the data that is actually used in the papers for evaluation or other purposes, with the same codes as the intended domain. This enables us to examine discrepancies between intended and actual domains.

**Quality criteria:** text properties practitioners focus on when evaluating summarization systems. We annotate this aspect to understand what is conceptualized as a "good" summary.

### 3.3.3 Limitations & Ethical Considerations

Lastly, we are also interested in both what kind of limitations (of both their work and prior work), ethical considerations, and stakeholders the authors explicitly bring up, as well as limitations that they might have overlooked.

**Limitations of prior work:** what authors describe as weaknesses of prior work to track what existing issues the authors engage with. To capture this, annotators copy or summarize passages where limitations of prior work are covered.

**Limitations of one's work:** whether and how the authors discuss the limitations of their own work. Annotators again either copy or summarize relevant passages.

**Other limitations identified by annotators:** the notes annotators took about any limitations they noticed while reviewing that were not already mentioned by the authors.

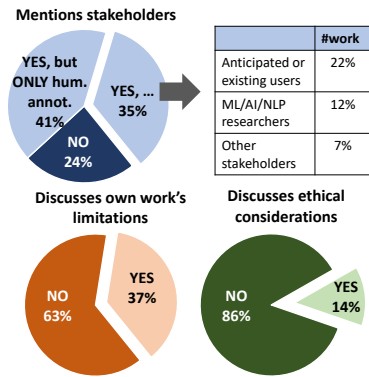

Figure 2: Summary statistics about mentioned stakeholders, and about how often papers cover limitations of one's work and ethical considerations.

| Contributions | # | Intended Domain | # |
|---|---|---|---|
| System | 224 | General | 173 |
| Dataset | 91 | News | 45 |
| Metric(s) | 36 | Dialogue | 38 |
| Evaluation | 73 | Opinion | 19 |
| Application(s) & Other | 34 | Medical | 17 |
| **Author Affiliation** | **#** | Scientific | 10 |
| Academic | 299 | Code | 9 |
| Industry | 121 | Wiki | 9 |
| (collab of above two) | (95) | Blog | 3 |
| Other | 32 | Other | 26 |

Table 1: Overview of the corpus of papers we reviewed.

**Ethical considerations:** whether there is a paragraph or section dedicated to ethical considerations, broader impacts of the work, or similar related topics, and if so, what is discussed therein. Annotators copy or summarize the section.

**Stakeholders:** whether the authors mention any stakeholders and who these stakeholders are, including *human annotators*, existing or anticipated *users* of a system, other *researchers* (in machine learning, AI, NLP, or related fields), or *other stakeholders*. We track this information because considering stakeholders is critical for envisioning harms and unintended consequences (Boyarskaya et al., 2020; Buçinca et al., 2023).

### 3.4 Analyzing Annotations

In addition to the codes we assigned to papers during the annotation, to further characterize particular practices (e.g., how many papers evaluate for factuality?), we also used keyword search and measured keyword frequency. To further assess various subsets of papers (e.g., papers that discuss their own limitations), we performed qualitative coding on extracted quotes, revisiting papers as necessary. Appendix B contain more details on this analysis (e.g., keywords used, codes for qualitative coding).

## 4 Findings

Our systematic review surfaced insights related to what the text summarization community has been focusing on (§4.1), common evaluation practices the community employs (§4.2), and how the community engages with ethical considerations (§4.3).

### 4.1 Community Focus

To help unpack why and how authors might (or might not) approach responsible AI considerations, we first wanted to understand who is conducting the research and how they conceptualize their work—what they are creating, who they are creating it for, and what outcomes they envision—helping us to contextualize current and possible future practices.

**There is a focus on developing new systems, with comparatively less emphasis on evaluation, metrics, datasets, or applications.** Nearly 70% of the 333 papers contribute new *systems* (including models and methods), while fewer than 30% contribute new datasets and around 10% contributed new metrics. An even smaller fraction (less than 2%) of papers focus on applications. This emphasis on developing new summarization systems, models, or techniques echoes concerns about the devaluation of e.g., data work which is often framed as "peripheral, rather than central" to AI research (Gero et al., 2023)—in contrast to the prestige of doing what is perceived to be more "technical" work such as modelling or system building.

**Many systems, metrics and datasets are intended to be general-purpose.** Assuming that not explicitly stating an intended domain means the work is implicitly intended to be "general-purpose," ~55% of 224 of papers contributing new systems intend them to be "general-purpose." Similarly, ~72% (26 out of 36) of papers contributing metrics and ~23% (21 out of 91) of those contributing datasets are also intended for general-purpose settings. However, these ostensibly general-purpose artifacts are often only tested or trained on a few domain-specific datasets and scenarios (§4.2).

**The text summarization literature remains driven by academic research,** though there is also significant interest from industry. ~90% of the reviewed papers were co-authored by at least one academia-affiliated author, with ~32% of these

papers being collaborations with industry authors. There are comparatively fewer papers solely written by authors affiliated with industry or other non-academic organizations. Because such organizations may be more connected to specific deployment settings and users (Zhou et al., 2022), their low representation may represent a barrier to engaging with the impacts of summarization systems.

**Papers rarely mention stakeholders when imagining intended use contexts.** While ~76% of reviewed papers mention stakeholders, fewer than half seem to mention stakeholders *other than* human annotators, who are typically mentioned in the context of evaluation practices rather than when discussing research goals. Only ~22% of all annotated papers consider anticipated or existing users, while ~12% mention other researchers. Conceptualizing a contribution without conceptualizing stakeholders may mean that the contribution will not meaningfully benefit any particular stakeholders, and may also make it more difficult to reason about limitations or adverse impacts.

**Papers referencing users are often both explicit about who those users are and specify an intended domain.** About three-fourths of the 74 papers we found to explicitly reference users, both describe who these users are and how they would benefit from automatic summarization, e.g., *"automatic summarizing tool that can generate abstracts for scientific research papers [...] can save much time for researchers and also readers"* (To et al., 2021). Many of these papers explicitly mention a specific intended domain, with only a small fraction of them (about 8%) (implicitly or explicitly) intending their work to be general-purpose. The remaining one-fourth only vaguely mention users, sometimes by specifying what users might want (e.g., *"a user might be looking for an overview summary or a more detailed one"* (Xu and Lapata, 2020)), without specifying who they might be. This is particularly the case when the authors intend for their work to be general purpose (67%, 12 out of 18). Not having a clear application or domain in mind can, however, make it difficult for authors to imagine users or other stakeholders.

**Imagined benefits often only include reducing anticipated users' labor or improving customer experiences.** ~54% (40 out of 74) of papers referencing users aim to reduce some type of labor. In these instances, the work is meant to automate, speed up, or even replace parts of users' workflow.

Examples include helping workers by summarizing meetings or emails (e.g., Singh et al., 2021; Zhang et al., 2022) and helping health professionals by summarizing medical encounters or files (e.g., Hu et al., 2022; Adams et al., 2021). ~20% of 74 papers referencing users aim to improve customer experience, which involves a transactional relationship between those who would deploy the summarization system and those who would use output summaries, for example, summarizing product reviews to *"make the shopping process more useful and enjoyable for customers"* (Oved and Levy, 2021) or summarizing livestreamed content to *"fully meet the needs of customers [on livestreaming platforms]"* (Cho et al., 2021). A more expansive conception of benefits might help practitioners consider more stakeholders, applications, and impacts.

## 4.2 Evaluation Practices

To examine current practices we considered actual domains (i.e., the domain of the data used or collected in the paper), researchers' conceptualization of summary quality, and which quality criteria they tend to prioritize.

**Most papers on general-purpose systems, metrics, and datasets *solely* use data from the *news* domain.** We estimate that ~52% of 122 papers contributing general-purpose systems only use *news* data when developing or evaluating systems, methods or models. This is not surprising since the most common summarization datasets are from the *news* domain (Dernoncourt et al., 2018; El-Kassas et al., 2021). For general-purpose metrics (i.e., not developed for only a restricted set of applications or domains and meant to be applied broadly), this percentage is ~77% out of 26 papers. Similarly, datasets introduced by papers that do not state an intended domain, or explicitly aim to be general-purpose, all collect their data from the *news* domain. These practices could introduce risks, as e.g., systems ostensibly developed to be general-purpose but only trained and evaluated on a restricted set of domains cannot be reliably used in other domains.

**While quality criteria concerning information saliency, linguistic properties, and factuality are frequently evaluated, criteria such as bias and usefulness are rarely evaluated, if ever.** Criteria related to information saliency (e.g., "informativeness," "relevance," "redundancy") are mentioned by ~41% of all reviewed papers. This is followed by criteria related to linguistic proper-

ties (e.g., "coherence," "fluency," "readability"), mentioned by ∼39% of papers, and criteria related to factuality (e.g., "factual consistency," "hallucination," "faithfulness"), mentioned by ∼28% of papers. Other criteria, such as summary usefulness (e.g., *"how useful is the extracted summary to satisfy the given goal, in our case, to answer the given query"* (Iskender et al., 2020)), and whether the summaries exhibit some bias (e.g., bias in text sentiment polarity (Sarkhel et al., 2020)) are rarely if ever mentioned. As a consequence, current practice seldom assesses whether more user-facing goals of summarization (e.g., the actual reduction of labor) are attained. Task-based evaluation, where summaries are assessed based on how they help humans perform a particular task (Lloret et al., 2018), is not a foreign concept in automatic summarization (Van Labeke et al., 2013; Zhu and Cimino, 2015; Jimeno-Yepes et al., 2013) and could be adopted by the community to better suit certain research goals.

**While factuality, information saliency, and linguistic properties are frequently evaluated, these criteria are less commonly conceptualized as part of research goals and limitations.** Comparatively, only ∼10% of all examined papers explicitly aim to address factuality-related qualities (e.g., better "evaluate faithfulness," "localizing factuality errors" in output summaries, or to prevent model "hallucinations"), and ∼15% of papers note factuality-related limitations in prior work (e.g., *"generating summaries that are faithful to the input is an unsolved problem"* (Aralikatte et al., 2021)). Similarly, only 8% of examined papers consider information saliency-related criteria as part of research goals, while 12% of papers point to these criteria when covering limitations of prior work. For linguistic properties, these percentages are 5% for research goals, respectively 9% for limitations of prior work. Naming these criteria as desirable, and explicitly targeting them in research, would facilitate the adoption of more careful operationalizations and engagement with the risks they may give rise to.

**Evaluation of output summaries still relies heavily on ROUGE** or on other similar automatic metrics based on lexical overlap, with ∼90% of 224 papers proposing new systems using such metrics. This fraction is ∼87% (79 out of 91) for papers contributing datasets and ∼70% (51 out of 73) for papers providing comprehensive evaluations. Overall, about 22% of all examined papers *only* use

these metrics. Since the reliability of ROUGE has been questioned (Novikova et al., 2017; Bhandari et al., 2020), there is a risk that metric scores do not reflect the true performance of evaluated systems.

**Human evaluation is widely used, but details on how it is carried out are often missing.** Some form of human evaluation seems used in a majority of papers, with ∼58% of all papers including mentions of human annotators. Yet our paper annotators noted limitations in how these evaluations were carried out for 24% (47 out of 194) of papers mentioning human annotators. Some of the issues most salient to our paper annotators included papers lacking detail about who the human evaluators are (noted for 22, ∼47% of 47 papers); the text properties or quality criteria human evaluators were asked to rate, such as asking annotators to score "importance" and "readability" without providing clear definitions (noted for 11, ∼23% of 47 papers); and the evaluation process in general, such as whether annotators were shown source documents during evaluation (noted for 9, ∼19% of 47 papers). These issues are particularly problematic for reproducibility and research standards. The community could adopt best practices developed for evaluation design, transparency, and analysis in human evaluation of text generation systems (van der Lee et al., 2019; Schoch et al., 2020).

## 4.3 Limitations and Ethical Considerations

Finally, we examine whether and how the community has engaged with ethical considerations and limitations of their own work and of existing work.

**Most papers do not discuss the limitations of their own work, and rarely include any ethical reflections.** We estimate that ∼63% of all annotated papers do not include a discussion about the limitations of their own work, while only ∼14% of surveyed papers have a section on ethical considerations. Papers proposing datasets are more likely to have an ethical considerations section (∼20%, 19 out of 92) than those proposing systems (∼10%, 23 out of 224). Work without such explicit reflections may not be able to effectively incorporate potential weaknesses or ethical concerns into the design and evaluation of their proposed systems, datasets, or metrics.

**When authors conceptualize ethical concerns, they often turn to data-related issues.** ∼62% of the 45 papers we found to include ethical considerations sections cover data issues in these sec-

tions. The data-related issues that are foregrounded include: data access and copyright (21 papers)—e.g., specifying that the data is publicly available; data privacy (13 papers)—mostly stakeholders who are either the people producing the data (e.g., professional writers of a scraped website (Liu et al., 2021)) or the people described by the data (e.g., users and customer service agents of e-commerce websites where data is collected (Lin et al., 2021)); and data "bias" (11 papers).

**When mentioned, data bias remains poorly defined or under-specified.** When discussing possible biases in their data, papers tend to only briefly and generically mention "bias" or a type of "bias" (e.g., "political bias", "gender bias", "biased views"). From our assessment, *only* 3 papers seem to provide more detail beyond these brief mentions (Adams et al., 2021; Cao and Wang, 2022; Zhong et al., 2021). Yet, even when bias issues are discussed in more depth, what is meant by data bias or the concerns or harms it can give rise to remain vaguely specified. For instance, (Zhong et al., 2021) mention how *"meeting datasets rarely contain any explicit gender information, [yet] annotators still tended to use 'he' as pronoun"* without further elaboration about e.g., the harmful stereotypes these biases might reproduce or whether the viewpoints of certain users might be unequally represented or misattributed in resulting systems' meeting notes summaries. While it is encouraging to see data bias identified as a source of concern, there is an opportunity to do so consistently and to provide a clearly articulated conceptualization of what is meant by data bias (Blodgett et al., 2020; Goldfarb-Tarrant et al., 2023).

**While papers often discuss limitations related to various quality criteria, these are rarely conceptualized as ethical concerns.** Papers describe a range of issues when reflecting on limitations. ~24% of the 122 papers we found to discuss limitations talk about factuality-related issues—ranging from only brief mentions (e.g., generic references to "factual errors" or "hallucinations"), to more detailed descriptions (e.g., *"factual errors by mixing up important details [such as] mixing up the victim and suspect of a crime, mixing up locations and dates"* (Panthaplackel et al., 2022)). Limitations related to linguistic properties (e.g., length, word novelty, coherence, fluency) are also sometimes mentioned (20 out of 122), as are issues related to information saliency or coverage (12 out of 122).

Of these issues, however, only factuality seems to be conceptualized as an ethical concern, with ~38% (17 out of 45) of papers with ethical consideration sections mentioning factuality-related concerns. From our assessment, no papers covering ethical concerns seem to relate them to quality criteria such as linguistic properties, or information saliency or coverage.

**While factuality is sometimes conceptualized as an ethical issue, few papers reflect on the impact of factual errors.** Only 6 of 17 papers (~35%) seem to name factuality as an ethical concern by describing adverse impacts of factuality-related model failures, with 4 naming "misinformation" or "bad influence" in the news domain, one "misinformation" in the context of corporate meetings (e.g., which *"would negatively affect comprehension and further decision making"* (Zhong et al., 2021)), and one the *"risk of misinterpretation of evidence and subsequent [medical] malpractice"* (Otmakhova et al., 2022).

The other 11 papers either generically describe factuality-related model failures (e.g., *"Even though our models yield factually consistent summaries [...] they can still generate factually inconsistent summaries or sometimes hallucinate information"* (Jiang et al., 2021), or describe factuality-related concerns as an "open problem" (Xiao et al., 2022) or as a problem with "unacceptable outcome" in "high-impact" domains (such as scientific and medical domains, DeYoung et al. (2021)) without much elaboration. A few papers (3) also explicitly warn that their models are not ready for deployment due to the lack of guarantees for the factual correctness of model outputs.

**Papers describing ethical considerations often do not engage with intended use context.** We estimate that fewer than half of the 45 papers explicitly considering ethical issues engage with intended use contexts. For example, only 3 papers explicitly mention the need for human oversight in system deployment, and only 2 of these describe the stakeholders who would be responsible for supervision with one paper noting that *"[t]he most natural application of this technology is not as a replacement for a human scribe, but as an assistant to one. By providing tools that aid a human scribe one can mitigate much of the risk of system failures, such as hallucination"* (Zhang et al., 2021a). Ethical considerations not grounded in use contexts may not be able to realistically anticipate adverse impacts.

**When stakeholders are mentioned in ethical considerations, potential harm to them is often overlooked.** Discussion of stakeholders is restricted to the compensation of human annotators (13 out of 45 papers), data privacy (13 out of 45 papers), and intended positive impacts on anticipated users (15 out of 45 papers). This may limit the conceptualization and evaluation of benefits and harms. The above requirement for human oversight, for instance, does not consider whether it might increase labor instead of reducing it, nor does it consider when or which stakeholders are well-equipped to supervise.

## 5 Discussion and Recommendations

**Intended use contexts are often not well-described.** We find that many papers do not specify an intended domain, and few works mention stakeholders, such as existing or intended users, when imagining intended use contexts. Even when such stakeholders are mentioned, imagined benefits are quite narrow in scope.

*Recommendations:* We encourage practitioners to conceptualize their contributions' intended use context by articulating, as much as possible, relevant stakeholders, intended domains, and potential benefits and adverse impacts to those stakeholders.

**Quality criteria such as bias and fairness are rarely considered.** We find that the priority in summarization evaluation is often on information saliency, linguistic properties, and factuality.

*Recommendations:* While these quality criteria are important, other criteria (e.g., social bias, usability) might be relevant and of interest to stakeholders. We encourage authors to consider these criteria, clearly define them (which may require grounding them in specific use contexts), and adopt evaluation practices that meaningfully capture them. To this end, we encourage the development of evaluation instruments (e.g., benchmarks or human evaluation protocols), especially those tailored for, or adaptable to, specific use contexts.

**There is a lack of engagement with limitations and ethical concerns.** We find that most papers do not have discussions on their own limitations, ethical considerations, and other related issues. When they do, they often focus on data-related concerns. This practice is not wrong by itself, but could be indicative of a narrow range of ethical concerns practitioners might be aware of. We especially

highlight two areas that tend to be overlooked: i) some model failures are rarely conceptualized as ethical concerns; ii) intended use contexts, including stakeholders, are rarely involved in ethical considerations, which prevents authors from imagining potential harm to said stakeholders.

*Recommendations:* To better engage with limitations and ethical concerns, we recommend to:

1) Reflect explicitly on the conceptualization of their work's intended use context, and of what constitutes a good summary in that context. What assumptions about system capabilities, stakeholders, or intended domains do choices of quality criteria and accompanying evaluations carry (Zhou et al., 2022)? What are the implications of these assumptions and choices?

2) Engage with prior literature (e.g., Bender, 2019; Weidinger et al., 2022) on ethical concerns and real harms to which NLP systems can give rise, such as hate speech, stereotyping, and misinformation. This could help practitioners critically reflect on their own work and more clearly engage with issues that have already been recognized as ethical concerns, such as "bias" (Blodgett et al., 2020).

3) Engage with ethical issues *the text summarization community* has already recognized. Through our survey we identified issues, such as the risk of misgendering stakeholders in summarization, which have already been pointed out by some members of the community e.g., (Zhong et al., 2021). Engaging with these issues could also help practitioners imagine limitations and ethical concerns of their own work.

## 6 Conclusion

We surveyed 333 recent text summarization papers from the ACL Anthology to examine how the summarization community currently conceptualizes and engages with broader responsible AI issues, and discuss how this might be impacted by existing research practices. While we are heartened by some of the practices we observed, such as evaluating issues like factuality, there remains significant opportunity to also foreground other responsible AI concerns. We hope that, by highlighting current practices and offering actionable guidance, this work will encourage a reflective, collective research and reporting practice in summarization research and beyond.

## 7 Limitations

Our findings are limited to the papers covered by our survey, which come from the ACL Anthology and are written in English. Works from other sources, such as venues with a different focus (e.g., venues focusing on AI applications) or those having a different demographic distribution than ACL, might paint a different picture. Our findings are also limited to the time period it covers: for example, between 2020-2022 some venues had not yet introduced the requirement to have a "Limitations" (or "Broader Impacts" or "Ethical Considerations") sections. We hope to see how the picture might change in the future as such requirements become more and more standard.

Furthermore, our findings are limited by our paper annotation process. The annotation guidelines described in Section 3.3 could overlook attributes which we failed to imagine with our current understanding of responsible AI issues and the task of automatic summarization. While we carried out the steps detailed in Section 3.2.2 with the goal of ensuring high annotation quality, the annotation process is imperfect; not all annotators are trained in responsible AI issues and they do not perfectly follow the annotation guidelines nor follow them the same way as a collective.

## 8 Ethical Considerations

As with any research undertaking, our work may also have unintended outcomes. By only foregrounding a subset of ethical and other responsible AI concerns in our discussion of the findings, we may inadvertently suggest that other issues deserve less consideration. The authors and annotators are also limited by our own conceptualizations of responsible AI issues, and there may be issues we fail to recognize.

## Acknowledgements

This work is supported by a joint Microsoft Research - Mila grant. Yu Lu Liu is also supported by a Fonds de Recherche du Québec Nature et Technologies master research scholarship (File #330991). Jackie C.K. Cheung is a consulting researcher at Microsoft Research Montréal. We thank the hired paper annotators for their contribution: Martin Pömsl, Sabina Elkins, Arjun Vaithilingam Sudhakar, Akshatha Arodi, Andrei Mircea, Kushal Arora, and Cesare Spinoso-Di Piano. We also thank Jules Barbe for his early work on this project. Finally, we thank the anonymous reviewers for their valuable feedback.

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

# A   Statistics on Paper Annotators

Annotators took on average around 23 minutes per paper. Excluding the exploratory review and Round 1, the number of papers coded by each annotator is:

**Round 2 totalling 131 papers**
- Author-Annotator 1: 15
- Author-Annotator 2: 20
- Author-Annotator 3: 5
- Author-Annotator 4: 2
- Author-Annotator 5: 84
- Author-Annotator 6: 5

**Round 3 totalling 142 papers**
- Hired Annotator 1: 12
- Hired Annotator 2: 14
- Hired Annotator 3: 21
- Hired Annotator 4: 42
- Hired Annotator 5: 42
- Hired Annotator 6: 2
- Hired Annotator 7: 9

While inter-annotator agreement (IAA) is often used as a proxy for annotation quality, particularly when trying to determine some ground truth, our paper reviews are not meant as a gold standard, and instead we looked for building consensus on ambiguous cases. We aim to surface issues in the current practices and provide rough estimates of how prevalent these issues might be. To ensure quality, we recruited annotators with expertise in the field, and encourage frequent discussions of ambiguous cases.

# B   Methodology

Here, we provide additional details about protocols we followed while coding and analyzing the set of papers included in our review.

| Code | Theme description | Num. Papers |
|------|-------------------|-------------|
| #annotator_pay | considerations related to how the annotators were paid | 13 |
| #annotator_description | demographic and other information related to the expertise of annotators | 5 |
| #data_access | how the data proposed/used can be accessed | 21 |
| #data_bias | concerns about biases in datasets | 12 |
| #stakeholder_privacy | privacy concerns related to stakeholders' information/data | 13 |
| #compute_info | concerns related to computational time and resources | 4 |
| #factuality | concerns related to factual errors, e.g.,"hallucinations" | 17 |
| #intended_use | describe the intended use domain, or the broader context | 21 |
| #no_deployment | explicitly mentions that deployment is too premature, not ready | 4 |

Table 2: Resulting codes and corresponding themes in "ethical considerations" sections.

| Code | Theme description | Num. Papers |
|------|-------------------|-------------|
| #novel_words | issues related to systems being unable to use words that are not present in the input text | 2 |
| #length | considerations related to the length of the input text or of the summary | 9 |
| #misgender | the output summaries referencing the wrong gender pronouns/terms | 3 |
| #relevance | whether the information contained in the output is relevant, non-redundant | 8 |
| #info_recall | whether key information from the source is included by the output | 7 |
| #factuality | aspects related to factual consistency, e.g.,"hallucinations" | 29 |
| #readability | properties related to e.g., coherence, fluency, grammar | 12 |
| #failure_mode | specific circumstances where model/metric/etc. perform poorly | 12 |
| #doubt_generalize | concerns about generalizability to other domains, languages, etc. | 13 |
| #weak_methods | known or potential weaknesses with their methodology | 29 |
| #weak_experiment | known or potential weaknesses with their experimental design | 26 |
| #complex_use | using a system or method is complex or requires extensive computational resources | 7 |

Table 3: Resulting codes and corresponding themes in authors' discussions of the limitations of their own work.

## B.1 Community Focus

To examine *research goals*, our analysis considered *mentioned stakeholders* as we were interested in how the authors envision anticipated or existing users to benefit from their work, and how these users are described. For this, we first identified the papers that mention users. As we observed that the code *other stakeholders* was sometimes used to denote users, we also manually filtered all papers coded *other stakeholders* for mentions of potential or existing users. When the description of research goals in these papers did not mention users, we revisited the papers to locate passages elaborating on how users benefit, which we then iteratively coded to identify the themes covered in Section 4.1.

To check whether commonly evaluated quality criteria such as factuality, information saliency, and linguistic properties were also conceptualized as part of research goals, we used the same keywords listed in the next section (§B.2) to estimate the number of papers focusing on these criteria (discussed in §4.2).

## B.2 Evaluation Practices

To surface insights about current evaluation practices, we primarily examined aspects related to the *actual domain*, as well as commonly considered *quality criteria*. For *actual domain*, we were interested in discrepancies with what the *intended*

*domain* was meant to be. For *quality criteria*, we examined the words authors frequently use to describe the quality criteria they consider, and performed the following keyword searches to estimate how often authors consider these criteria:

– information coverage: "relevan" (for relevant/relevance), "repetition", "informat" (for information/informativeness), "redundancy", "salien" (for salient/saliency), and "content coverage"

– information presentation: "fluen" (for fluent, fluency), "gramma" (for grammar, grammaticality), "readab" (for readable, readability), "coheren" (for coherent, coherence), "length", "novel" (for novel words)

– factuality: "factual" (also for factuality), hallucinat (for hallucinate, hallucination), faithful (also for faithfulness), consisten (for consistency), correct (also for correctness).

To estimate how frequently ROUGE-like automatic metrics are used, was tracked it using the tag "ROUGE" during the paper annotations.

## B.3 Ethical Considerations

After inspecting the annotators' summaries provided by the annotators for the ethical consideration sections, we discarded 2 papers which we found to be mistakenly annotated as having an ethical consideration section: i) Krishna et al. (2021): the annotation pulled passages from the abstract.

There's no ethical consideration section in the paper. ii) Mullenbach et al. (2021): the paper has a "potential impact" section in the introduction that we believe addresses the paper goal.

The specific codes we obtained after iteratively coding the ethical considerations sections to surface themes are listed in Table 2.

### B.4 Limitations of one's own work

The specific codes we obtained after iteratively coding the passages about the authors discussions of the limitations of their own work are listed in Table 3.