# OpenReview forum: "Responsible AI Considerations in Text Summarization Research: A Review of Current Practices"
_EMNLP/2023/Conference — EMNLP 2023 Findings_

### Official Review · Reviewer_8kcV · 2023-08-02

**Typos Grammar Style And Presentation Improvements:** 1. In line 534, please include the mi…
**Soundness:** 3

**Excitement:**

3: Ambivalent: It has merits (e.g., it reports state-of-the-art results, the idea is nice), but there are key weaknesses (e.g., it describes incremental work), and it can significantly benefit from another round of revision. However, I won't object to accepting it if my co-reviewers champion it.

**Paper Topic And Main Contributions:**

This paper provides a meta-analysis of Text Summarization Papers, focusing on their alignment with responsible AI considerations. By examining how well research papers on text summarization address ethical and responsible AI practices, the study sheds light on the field's current state and potential implications for developing more responsible AI models.

**Questions For The Authors:**

I have the following questions regarding the paper:
1. Can you provide an explanation of how the three findings mentioned in the Conclusion (Section 5) are related to the findings presented in Section 4?
2. Have you conducted an analysis of the usability of the methods described in the papers you examined? For instance, did you investigate how easily the results of these methods can be reproduced, or whether any of these methods have adverse effects on the environment?

**Reasons To Accept:**

There are two main reasons to accept the paper:

1. **Creation of Annotated Dataset**: The authors have meticulously developed an annotated dataset containing summarization papers from ACL Anthology spanning the years 2020 to 2022. This dataset includes essential information such as paper goals, authors, evaluation practices, limitations, and ethical considerations. This comprehensive resource could facilitate further analysis in this domain enabling exploration into how text summarization has been influenced by other NLP tasks and how recent technological advancements have impacted the field.

2. **Systematic Summarization and Clear Structure**: The paper presents a systematic summary of all the research findings, supported by statistical evidence. Moreover, the authors have thoughtfully organized the findings, enhancing the overall clarity and coherence of the paper.

**Reasons To Reject:**

There are three primary reasons to reject the paper:

1. **Limited Analysis Scope**: The paper only focuses on analyzing the metadata of the papers, neglecting valuable information present in the contents of these papers. A more comprehensive analysis that incorporates the actual content could have strengthened the presented findings with additional evidence and insights.

2. **Superficial Analysis and Weak Evidence**: The analysis conducted in the paper is deemed superficial by me (I agree that this opinion is subjective), lacking sufficient strong evidence to support the claims made. For instance, at line 571, a claim is stated without concrete examples from the corpus to substantiate it ("Work without such explicit reflections may not be ....."). Including representative examples could have provided more robust evidence for the conclusions.

3. **Lack of Clarity in Findings**: The presented findings do not clearly explain the direction of research in text summarization. The paper appears to fall short in effectively conveying the implications and significance of the results.

**Reproducibility:**

N/A: Doesn't apply, since the paper does not include empirical results.

**Reviewer Confidence:**

3: Pretty sure, but there's a chance I missed something. Although I have a good feel for this area in general, I did not carefully check the paper's details, e.g., the math, experimental design, or novelty.

---

> ### Author Rebuttal · Authors · 2023-08-29
>
> We thank the reviewer for their feedback and will address their general concerns, followed by answers to their questions.
>
> # General Concerns
> ## Analysis Scope
> We actually looked at both metadata AND content of each annotated paper. Many aspects in our annotation scheme concern the content of the paper: how the paper authors discuss limitations of their own work, the stakeholders they mention (if any), what they discuss within their paper’s ethical consideration section (if there is such a section), etc. Using a paper’s metadata alone, this valuable information is unobtainable.
>
> ## Representative Examples
> We agree that including more representative examples will strengthen our findings. To improve the current version, we will add more quotes to illustrate a certain practice some of the annotated papers are doing (e.g., in lines 613-616, we quoted from Panthaplackel et al., 2022 to illustrate specific descriptions of factuality-related issues).
>
> ## Clarity in Findings
> We thank the reviewer for pointing out this problem to us. We now realize that our organization of the concluding paragraphs (recommendations in Section 4.4, followed by the summary of findings in the Conclusion) has caused confusion. We will re-organize these paragraphs and clarify the implications and significance of the results by creating a new “Discussion” section where we will better summarize the findings, offer our recommendations, and better link the two.
>
> # Questions
> ### Q1: link between findings and conclusion
> The above point leads us to answer the reviewer’s Q1: The three points presented in the conclusion were meant to summarize the findings in Section 4 (except for recommendations in 4.4). The following improved structure might offer more clarity:
>
> **A. Intended use contexts are often not well-described** (currently conveyed in conclusion (i)). We view intended use contexts through 2 lenses:
> - Intended domain of application (i.e., where the work is intended to be used): There are many works that do not have a specific domain of application, implicitly intended to be “general-purpose” (Ln 387).
> - Mentioned stakeholders (i.e., who are the intended users or people impacted by the work): few works mention stakeholders when imagining intended use contexts (Ln 406). Imagined benefits are also quite narrow in scope (Ln 444).
>
> **B. Quality criteria such as bias, fairness, etc., are rarely considered.** (currently conveyed in conclusion (iii)). The priority in evaluation is on information saliency, linguistic properties, and factuality (Ln 486).
>
> **C. We see a lack of engagement with limitations of own work and ethical concerns** (currently partially conveyed in conclusion (i) and (ii)):
> - Most works do not have discussions on their own limitations, ethical considerations, etc. (Ln 562). Work that do often focus on data-related concerns (which is not wrong by itself, but could be indicative of a narrow range of ethical concerns practitioners might be aware of)
> - Model failures (e.g., factual errors) are rarely conceptualized as ethical concerns. (Ln 605, Ln 627)
> - When considering ethical considerations, intended use contexts (including stakeholders) are rarely involved. Potential harm to stakeholders are thus often overlooked (Ln 653, Ln 673)
>
> ### Q2: on analyzing the usability, reproducibility, impact on environment, etc. of the methods
> No, we did not conduct such an analysis. Our study did not examine properties of the methods such as how reproducible they are, or their impact on the environment. Rather, we focus on how authors engage with these aspects, and potential RAI issues that could arise. To illustrate, instead of investigating how much computational resources each annotated paper actually used (i.e, not the focus of the paper), we noted, in Appendix C.2.3, that 4 papers described their use of computational resources when discussing limitations of their own work (i.e, the authors of these papers *engaged* with the usage of computational resources *when imagining limitations of their own work*).
>
> Finally, we thank the reviewer for their suggestion on including visualization. We will consider doing so in our revised version if space permits. We will also fix the missing number on Ln 534: it should read 200 of 224.

---

### Official Review · Reviewer_ao4U · 2023-08-02

**Soundness:** 2

**Excitement:**

3: Ambivalent: It has merits (e.g., it reports state-of-the-art results, the idea is nice), but there are key weaknesses (e.g., it describes incremental work), and it can significantly benefit from another round of revision. However, I won't object to accepting it if my co-reviewers champion it.

**Paper Topic And Main Contributions:**

This paper provides a systematic review on responsible AI practices within the subfield of summarization. The authors start with an extensive annotation procedure where they look more specifically at (1) paper authors & goals, (2) data & evaluation practices, and (3) limitations & responsible AI. The main goal of the paper is to investigate to what degree the implications of summarization work is reported on. The authors find that this is currently not being done thoroughly.

**Questions For The Authors:**

The authors conduct an interesting review on the current state of text summarization. They go through an extensive annotation process, making their results reliable (although I do have remarks on this at a later stage).

However, my main gripe with this paper is that because the review is set up so broadly, a main point or goal seems to be lacking. The paper is set up as to focus on responsible AI, which is only a small part of their systematic review. The authors discuss various points related to paper authors & goals, data & evaluation practices, besides limitations & responsible AI. Therefore, it is difficult to find the main narrative in the findings: for instance, how does evaluation correspond with ethics, and stakeholder mentions? Is it a responsible AI issue that papers rarely mention stakeholders (and why is this)? What does it mean for responsible AI that work in summarization is largely driven by academic research? All these links are left implicit throughout the paper, so that it is difficult for readers to take away lessons and apply the findings to their own research.

For instance, when I read passages such as "indeed, not having a clear application or domain in mind can make it difficult for authors to imagine users or other stakeholders" I quite desired an extra step here which explicitly link this to responsible AI, by mentioning something like "which, in turn, makes it more challenging to describe in what ways their work can be potentially harmful."

Furthermore, I am missing an argument from literature that describes why for instance factuality can be an ethical issue, or what kind of harms to stakeholders can occur. Explaining the relevance and prevalence of certain ethical AI issues will also help to provide researchers with starting points to better their responsible AI considerations. For instance, statements such as "despite the very real harms to which
NLP systems can give rise (e.g., Bender, 2019; Weidinger et al., 2022)" warrant further explanation on the very real harms.

I will address this more specifically in a question format, and other questions related to clarity and methodological issues below.

- General (see point made above): To what degree do the different themes from the systematic review relate to responsible AI issues?
- General, but more specifically 131-134 "While a great deal of work on responsible AI issues has emerged for NLP broadly, much less work has addressed summarization specifically." --> Why does summarization specifically need attention in this regard? How are developments there different from what is happening with NLP/NLG in general?
- General (but mostly related to §4.3 and §4.4, see point made above) --> Could you provide more information on the relevance and prevalence of certain ethical/responsible AI issues?
- 197-198: "e.g., NLG papers where summarization is one of many evaluated tasks" --> Does this also include papers that are not experimental in nature, such as papers discussing evaluation practices of summarization? This is addressed later on, but it would be good to make explicit that papers do not necessarily have to be experimental in nature here.
- §3.2.1 --> Can you give details on the number of annotators and details on the annotators who did this exploratory review? Were these all the authors of this paper?
- §3.2 --> Do you have any notes on the total time it took annotators to go through their batches? Any details on how many papers were coded by multiple annotators? Any IAA metrics?
- 289 "or when the paper explicitly intends to be general" --> What is meant by this? Applicable to multiple domains?
- §3.3.1 Research goals --> Could no taxonomy be made here?
- §3.3.1 Research goals --> Could you give a brief explanation why this category is relevant (in the context of responsible AI)?
- §3.3.1 Quality criteria --> What kind of taxonomy is used here? Is specific distinction made between automatic and human evaluation?
- 583-588 " data privacy (13 papers)—most specifying the stakeholders, who are either the people producing the data (e.g., professional writers of a scraped website) or the people described by the data (e.g., users of e-commerce websites where data is collected)" --> In what way is privacy here related to the stakeholders most commonly?

**Reasons To Accept:**

- Thorough systematic review, with a well-described annotation process
- Interesting results for the field

**Reasons To Reject:**

- Paper lacks coherence: connection between the results and the conclusion for the field are missing
- Narrow topic (summarization)
- Background information on why responsible AI issues are actually such issues is often left implicit.

**Reproducibility:**

4: Could mostly reproduce the results, but there may be some variation because of sample variance or minor variations in their interpretation of the protocol or method.

**Reviewer Confidence:**

5: Positive that my evaluation is correct. I read the paper very carefully and I am very familiar with related work.

**Typos Grammar Style And Presentation Improvements:**

- 053 "RAI" --> This abbreviation needs to be defined first.
- 128-130 "Our work examines how assessment practices might limit the space of responsible AI issues the community considers." --> As a reader, I find it hard to understand what is meant with "assessment practices" and "limit the space of responsible AI issues". Could you reformulate this?
- 385 "more than half" --> Can you give an exact percentage?
- 530-532 "Linguistic properties and criteria related to information saliency are, in comparison, less frequently mentioned" --> Could you specify how often they are mentioned?
- 570-571 "(∼20% (19 out of 92) than those proposing systems (∼10% (23 out of 224))" --> This is somewhat of a personal preference, but try to avoid nested parentheses.

---

> ### Author Rebuttal · Authors · 2023-08-29
>
> We appreciate the reviewer’s thoughtful and detailed comments. We will first respond to the reviewer’s general concerns, then answer the more specific questions, and finally answer those related to style and presentation.
>
> # General Concerns
> ## Our Focus on the Task of Summarization
> We respectfully disagree that summarization is a narrow topic. Not only is there a rich literature on summarization (El-Kassas et al., 2021., Widyassari et al., 2022., etc.), but it is also one of the tracks at EMNLP this year.
>
> This brings us to answer the reviewer's question: **“Why does summarization specifically need attention in this regard? How are developments there different from what is happening with NLP/NLG in general?”**
>
> Although findings from work on RAI in NLP/NLG could apply to summarization, there are task specificities that may be neglected (perhaps for good reasons, as these work aims for a much wider scope) that deserve their own examination. For example, summarizing necessarily involves omitting less relevant information from the source document -- unlike translation, for instance. Whether summarization systems fairly reflect different opinions or contributions from different authors is thus worth investigating (Carenini and Cheung (2008), Shandilya et al. (2018), Dash et al. (2019) which are cited in our related work section).
>
> In the same way, due to these task specificities, there may be practices and RAI issues that are specific to summarization and are worth studying as they are not addressed in previous work. Other NLP/NLG tasks also have their own specificities and are also worth studying -- we hope that our methodology could be adapted to study, for instance, how practitioners engage with RAI issues in machine translation, or in dialogue generation, etc.
>
> ## Background on Responsible AI Issues
> We thank the reviewer for raising the concern that readers may not be familiar with responsible AI issues. We will add explanations and relevant literature to make it easier for readers to understand the rationale behind the aspects covered by our annotation scheme and the implications of the issues we identify. We will briefly outline them here, thereby also answer the reviewer’s first general question:
>
> - **Paper authors & goals** contextualize our findings. As we aim to examine how practitioners engage with RAI in their work, we need to know who are the practitioners (author affiliation), what is the work (type of contribution), and what are their intentions behind their work (research goals, intended domain). This also provides cues to potential usage scenarios, which may determine what harms are likely to occur.
> - **Evaluation practices** could both limit the space of RAI issues the community is aware of (further elaborated in the response to one of the reviewer’s clarification questions) and lead to RAI issues. For example, evaluating “general-purpose” systems on only news domain data could lead to practitioners being unaware of how said systems behave in other domains of application: do they perform reliably and safely?
> - As for **stakeholders**, the reviewer is right: not considering stakeholders makes it very challenging to think about potential harms (e.g., trying to imagine the harm of summarization systems in the medical domain without thinking about patients).
>
> Another related point is about the **relevance and prevalence of certain issues** (also presented as the 3rd general question from the reviewer). We thank the reviewer for bringing it to our attention, and we will clarify the link between issues identified in our work and their relevance/prevalence in the broader literature.
>
> For factuality-related ethical issues, for example, we will provide pointers to literature on the impacts of misinformation (e.g., Loomba et al., 2021 on the impact of misinformation on health) and lack of factuality in general or in domains like news (e.g., Wilner et al., 2022 on the impact of factual errors on media trust).
>
> The importance of many ethical issues are currently explained in the cited works. For example, Weidinger et al., 2022 presents a taxonomy of risks covering discrimination, harmful stereotypes, privacy, impacts of false information, malicious use, etc. As suggested by the reviewer, we will draw clearer links between the work we cite and the issues we describe in text by providing as much explanation as we can in text.
>
> ## Connection between Findings and Conclusion for the Field
> We thank the reviewer for pointing out this problem to us. We now realize that our organization of the concluding paragraphs (recommendations in Section 4.4, followed by the summary of findings in Conclusion) has caused confusion. We will re-organize these paragraphs and clarify the implications and significance of the results by creating a new “Discussion” section where we will better summarize the findings, offer our recommendations, and better link the two:
>
> **A. Intended use contexts are often not well-described** (currently conveyed in conclusion (i)). We view intended use contexts through 2 lenses:
> - Intended domain of application (i.e., where the work is intended to be used): There are many works that do not have a specific domain of application, implicitly intended to be “general-purpose” (Ln 387).
> - Mentioned stakeholders (i.e., who are the intended users or people impacted by the work): few works mention stakeholders when imagining intended use contexts (Ln 406). Imagined benefits are also quite narrow in scope (Ln 444).
>
> **B. Quality criteria such as bias, fairness, etc., are rarely considered.** (currently conveyed in conclusion (iii)). The priority in evaluation is on information saliency, linguistic properties, and factuality (Ln 486).
>
> **C. We see a lack of engagement with limitations of own work and ethical concerns** (currently partially conveyed in conclusion (i) and (ii)):
> - Most works do not have discussions on their own limitations, ethical considerations, etc. (Ln 562). Work that do often focus on data-related concerns (which is not wrong by itself, but could be indicative of a narrow range of ethical concerns practitioners might be aware of)
> - Model failures (e.g., factual errors) are rarely conceptualized as ethical concerns. (Ln 605, Ln 627)
> - When considering ethical considerations, intended use contexts (including stakeholders) are rarely involved. Potential harm to stakeholders are thus often overlooked (Ln 653, Ln 673)
>
> With these 3 points presented first, it would be hopefully easier to understand where our recommendations comes from:
> - Our 1st recommendation (try conceptualizing the intended use context of a contribution) is directly connected to point A above.
> - Our second recommendation (being explicit about how a “good” summary is conceptualized) is directly connected to point B above and to our findings about domain mismatch in evaluation (Ln 473), reliance on ROUGE and similar metrics (Ln 533), and underspecified human evaluation (Ln 545).
> - Remaining recommendations about engagement with ethical concerns are all tied to point C above.
>
> # Specific Questions
> ### Ln 197-198 papers that are not experimental in nature
> No, we did not filter collected papers based on whether they are experimental in nature or not. We will follow the reviewer’s suggestion and emphasize this. Here, we aimed to exclude papers where summarization is not the main focus. Using the reviewer’s example, a paper discussing evaluation practices in summarization would be included whereas a paper discussing evaluation practices of NLG (covering other tasks such as machine translation, question generation, etc.) would be filtered out.
>
> ### §3.2.1: details on the exploratory review
> The exploratory review is indeed done by the authors of this paper, a team of 6 people. We unfortunately cannot provide further details on our experience/background due to concerns about anonymity.
>
> ### §3.2: details on annotation time, number of papers/annotator, inter-annotator agreement
> Once the annotation scheme was finalized, annotators took on average around 23 minutes per paper. Excluding the exploratory review and Round 1, the number of papers each annotator coded is as follows:
>
> (Round 2 totalling 131 papers)
> - Author-Annotator 1: 15
> - Author-Annotator 2: 20
> - Author-Annotator 3: 5
> - Author-Annotator 4: 2
> - Author-Annotator 5: 84
> - Author-Annotator 6: 5
>
> (Round 3 totalling 142 papers)
> - Hired Annotator 1: 12
> - Hired Annotator 2: 14
> - Hired Annotator 3: 21
> - Hired Annotator 4: 42
> - Hired Annotator 5: 42
> - Hired Annotator 6: 2
> - Hired Annotator 7: 9
>
> As for inter-annotator agreement, we understand that it is often used as a proxy for annotation quality, particularly when trying to determine some ground truth. Our annotations are, however, not meant as gold standard; rather, the goal of our annotation process is to surface issues in the current practice and provide estimates of how prevalent these issues might be. We thank the reviewer for drawing our attention to these statistics. We will include them in the appendix.
>
> ### Ln 289: papers explicitly intending to be general
> When a paper explicitly intends to be general, it explicitly states that its contribution is general-purpose, and that it can be used in any domain or application. We will add a sentence clarifying this.
>
> ### §3.3.1: research goals
> We did not make a taxonomy for research goals because authors describe their goals in many different ways; we sought to capture these descriptions as faithfully/directly as possible, and thus kept this category open-ended. We address the second question on research goals in our response to the reviewer’s general concerns above.
>
> ### §3.3.2: quality criteria
> We did not use a taxonomy when annotating for “quality criteria”, nor did we make a distinction between automatic and human evaluation; like the “research goals” category, this category was kept open-ended, where annotators extracted text properties that papers mention when describing summarization systems’ evaluation.
>
> ### Ln 583-588: data privacy and stakeholders
> We are not sure we understood the reviewer's question, but we will try to answer it by clarifying this passage. We meant here that when data privacy is brought up as an ethical concern, the authors often specify the people who could be impacted: people producing the data and people described by the data (the mentioned stakeholders). We could imagine, for instance, that a dataset that is not properly anonymized could leak private information of these stakeholders, thereby harming them. We hope this clarification answers the reviewer's question. If not, we ask the reviewer to please elaborate and we will answer in our future reply.
>
> # Questions Related to Style and Presentation
> We thank the reviewer for identifying the “RAI” abbreviation issue and nested parentheses, and we will fix them.
> We also thank the reviewer for identifying instances where we should have presented exact numbers and percentages. We will do so in our revised version:
> - Ln 385: 54.5% → 122 out of 224 papers contributing new systems.
> - Ln 530-532:
>     - Factuality Related: 32 in goals, 46 in prior work limitations;
>     - Information saliency: 27 in goals, 44 in prior work limitations;
>     - Linguistic properties (e.g., fluency, novel words, length, etc.): 17 in goals, 29 in prior work limitations;
>
> Finally, to clarify **Ln 128-130**, we meant that it is possible that some evaluation practices (e.g., the use of ROUGE, evaluating for factual consistency, etc.) might lead to some RAI issues being considered while others might be overlooked by the community (i.e. “failure of imagination”). To illustrate, if not evaluating for a certain property (e.g., factuality) is a common evaluation practice, then model failures related to said property (e.g., factual errors in output summaries) would likely be overlooked as a potential cause of harm. If only using news domain data is a common practice, then the community might fail to imagine how summarization systems could be used/misused and cause harm in other domains of application. We will reformulate this part to make this point clear to readers.
>
> # References
> Wafaa S. El-Kassas, Cherif R. Salama, Ahmed A. Rafea, and Hoda K. Mohamed. 2021. Automatic text summarization: A comprehensive survey. *Expert Systems with Applications*, 165:113679.
>
> Sahil Loomba, Alexandre de Figueiredo, Simon J. Piatek, Kristen de Graaf, and Heidi J. Larson. 2021. Measuring the impact of covid-19 vaccine misinformation on vaccination intent in the uk and usa. *Nature Human Behaviour*, 5(3):337–348.
>
> Adhika Pramita Widyassari, Supriadi Rustad, Guruh Fajar Shidik, Edi Noersasongko, Abdul Syukur, Affandy Affandy, and De Rosal Ignatius Moses Setiadi. 2022. Review of automatic text summarization techniques methods. *Journal of King Saud University - Computer and Information Sciences*, 34(4):1029–1046.
>
> Tamar Wilner, Ryan Wallace, Ivan Lacasa-Mas, and
> Emily Goldstein. 2022. The tragedy of errors: Political ideology, perceived journalistic quality, and media trust. *Journalism Practice*, 16(8):1673–1694.

---

### Official Review · Reviewer_vqro · 2023-08-03

**Soundness:** 4

**Excitement:**

4: Strong: This paper deepens the understanding of some phenomenon or lowers the barriers to an existing research direction.

**Paper Topic And Main Contributions:**

This paper presents a collection, annotation, and analysis of about 330 summarization papers published 2020-2022 with a focus on responsible AI. It describes an initial scoping annotation by the authors, resulting in an annotation scheme covering a variety of aspects (contributions, analysis, measurements, stated intent, and interactions with potential users). This annotation is used to identify several (sometimes implicitly bad) themes in summarization research, and the paper makes several explicit recommendations for performing summarization research.


**Questions For The Authors:**

(A) Did this study include workshops affiliated with main conferences? Shared tasks are a common feature of many workshops, and those are often explicitly built are system building.


**Reasons To Accept:**

The paper is very well written and easy to follow. It contains a high-quality analysis of recent summarization trends in the *ACL community, and highlights some commonalities of the research approaches. It makes some reasonable recommendations with respect to research focus and aligning researchers with their intended users, and finds that researchers are often not aligned.

**Reasons To Reject:**

(A) I question the utility of this work at this time - (1) system performance has only recently reached performance levels that we, as an academic community, need to care about metrics beyond ROUGE (and that does not seem to have any impact on what industry does), (2) on any meaningful time-scale, the community has only recently started to prioritize ethics and limitations considerations. I am unaware of any longer-term study of the effectiveness, impact, or other longer term effects of these new section requirements on paper quality, content discussed, or informativeness of that content.
I don’t see this as a true reason to reject this paper, but an incompletely analyzed aspect that should be considered when making research recommendations (are we already trending in the correct direction?).

(B) There are no measures of annotation quality or agreement. The paper explicitly calls out papers for lacking details on human evaluation (line 545-) but fails to provide any quality measures of the evaluation process results.b


**Reproducibility:**

4: Could mostly reproduce the results, but there may be some variation because of sample variance or minor variations in their interpretation of the protocol or method.

**Reviewer Confidence:**

4: Quite sure. I tried to check the important points carefully. It's unlikely, though conceivable, that I missed something that should affect my ratings.

**Typos Grammar Style And Presentation Improvements:**

Major:
Line 366-380 “There is a focus on developing new systems, with comparatively…”: Alternatively the higher number of papers may be because developing a new model is simply easier than collecting quality data. Or modeling may have more money or (arbitrarily) news coverage. By not considering alternative explanations, this paper chooses a particular bias in understanding why certain work gets done. The models all use data-if we use citations (a crude measure of importance), are there more data citations, or are there more modeling citations in the field? Is the comparatively small number of metric development papers a sign that metrics are considered important, or more a sign that we (as a field) lack good metrics? My own views directly contradict the supposition in this paper: data work is more enduring and impactful than any particular system, and system work is in some sense disposable when a higher quality system arrives.

Nits:
Line 53: RAI is used before definition.
Line 65: about 333?

---

> ### Author Rebuttal · Authors · 2023-08-29
>
> We thank the reviewer for their comments and we are grateful that they are recognizing the contributions of our work. We will first address their general concerns, followed by answers to their questions.
>
> # Utility of Our Work
> We first address the reviewer’s doubt about the work’s utility. We are somewhat confused by the reviewer’s comment (1) about **system performance and use of metrics**. The academic community has cared about metrics beyond ROUGE: prior work -- dating many years back -- has questioned the use of metrics like ROUGE (e.g., Murray et al., 2005) and proposed alternatives (e.g., Hovy et al., 2006). As for industry practitioners, as they need to deal with user facing settings, they often have to go beyond metrics like ROUGE (e.g., Zhou et al., 2022). We would appreciate it if the reviewer can clarify this comment and elaborate on their thoughts about the connection between the performance of current text summarization systems and the utility of our work.
>
> As for the comment (2) about the **timing**, the reviewer is correct: NLP conferences have only recently started to encourage (and later require) ethical considerations, limitations, and other related sections; and thus these practices are fairly new. We believe it is helpful to take an early snapshot of emerging practices in the community. While the community has only recently started to prioritize these issues, we believe our work helps us understand, early on, why the community might be struggling with considering limitations of its work, ethical issues, etc.
>
> # Annotation Quality
> About annotation quality, we did not report agreement measures as our annotation was done via open-coding, meant to surface issues in the current practice and provide estimates of how prevalent these issues might be. We ensured annotation quality via selection of annotators (expertise in the field) and via frequent meetings to discuss ambiguous cases. We also provided details on the annotators and the annotation process in Section 3.2.2.
>
> # Questions
> ### On whether workshop papers are included
> We thank the reviewer for pointing this out to us! We did not include workshops affiliated with the main conference in our review. Not including this information was an oversight on our side. We will clarify this in the paper and add a note about it when discussing the limitations of our work.
>
> ### Line 366-380 prevalence of system work
> We thank the reviewer for raising this concern. The statistics we present are actually not about the impact or the importance of each type of contribution, but rather about their prevalence in the papers we reviewed. To avoid misunderstandings, we will clarify this by rephrasing the sentence at line 366: “Papers contributing new systems are prevalent, whereas comparatively less papers work on evaluation, metrics, datasets, or applications.”
>
> We actually believe lines 373-380 are in line with the reviewer's view about the devaluation of data work, which we wholeheartedly agree with:
> > “This emphasis on developing new summarization systems, model, or techniques echoes concerns about the devaluation of e.g., data work which is often framed as ‘peripheral, rather than central’ to AI research (Gero et al., 2013). -- in contrast to the prestige of what is perceived to be more ‘technical’ work such as modelling or system building.”
>
> Finally, we thank the reviewer for identifying mistakes relating to style and presentation: we will define RAI before using this abbreviation and will delete “about” on line 65 as we annotated exactly 333 papers.
>
> # **References**
> Eduard Hovy, Chin-Yew Lin, Liang Zhou, and Junichi Fukumoto. 2006. Automated summarization evaluation with basic elements. In *Proceedings of the Fifth International Conference on Language Resources and Evaluation (LREC’06)*, Genoa, Italy. European Language Resources Association (ELRA)
>
> Gabriel Murray, Steve Renals, and Jezation Evaluation
> with Basic Elements. Carletta. 2006. Evaluating automaticed summaries of meeting recordings. In *Proceedings of the ACL Workshop on Intrinsic and Extrinsic EFifth International Conference on Language Resources and Evaluation (LREC’06)*, Genoa, Italy. European Language Resources Association (ELRA).
>
> Kaitlyn Zhou, Su Lin Blodgett, Adam Trischler, Hal Daumé III, Kaheer Suleman, and Alexandra Olteanu. 2022. Deconstructing NLG evaluation: Evaluation practices, assumptions, and their implications. In *Proceedings of the 2022 Conference of the North American Chapter of the Association for Computational Linguistics: Human Language Technologies*, pages 314–324, Seattle, United States. Association for Computational Linguistics.

---

### Meta-Review · Area_Chair_nMxT · 2023-09-19

**Recommendation:** 3

**Metareview:**

The paper "Responsible AI Considerations in Text Summarization Research: A Review of Current Practices" presents a survey of how ethical issues are dealt with in the context of automatic summarization.

The main criticism voiced by the reviews refers to the narrow scope of the paper and the lack in connection between aspects of the paper.
Points in favour of the paper are the details of the survey and the data set created for this paper.

---

### Decision · Program_Chairs · 2023-10-07

**Decision:**

Accept-Findings

**Comment:**

The paper "Responsible AI Considerations in Text Summarization Research: A Review of Current Practices" presents a survey of how ethical issues are dealt with in the context of automatic summarization.

The main criticism voiced by the reviews refers to the narrow scope of the paper and the lack in connection between aspects of the paper.
Points in favour of the paper are the details of the survey and the data set created for this paper.